# Trends in Human Immunodeficiency Virus-Related Knowledge and Stigma among Men Who Have Sex with Men in the Republic of Korea from 2012 to 2022

**DOI:** 10.3390/healthcare11243135

**Published:** 2023-12-11

**Authors:** Aeree Sohn

**Affiliations:** Department of Public Health, Sahmyook University, Seoul 01795, Republic of Korea; aeree@syu.ac.kr; Tel.: +82-10-3399-1669

**Keywords:** HIV/AIDS, knowledge, stigma, MSM

## Abstract

To evaluate the trends in human immunodeficiency virus (HIV)-related knowledge and stigma among men who have sex with men (MSM) in the Republic of Korea between 2012 and 2022, five cross-sectional surveys were conducted. Though general knowledge of HIV infections is high, some misconceptions persist. The initial set of five questions in the survey maintained consistent levels of understanding throughout the five recurring surveys. Notably, the study found a consistent decrease in personal stigma toward people with HIV/AIDS (PWHA) within the MSM community in Korea. The increasing willingness of individuals to engage in social interaction with HIV-positive individuals, ranging from dining to cohabitation, indicated a positive change in societal attitudes. It is crucial to implement active policies such as counseling, testing, education, promotion, and the creation of preventive programs to improve knowledge about HIV and reduce stigma.

## 1. Introduction

Human immunodeficiency virus infection and acquired immune deficiency syndrome (HIV/AIDS) remains a global public health concern, notwithstanding leaps in medical understanding and therapeutic advancements (Cho, 2016 #66; UNAIDS, 2020 #228) [1,2]. Korea, which has a comparatively low HIV prevalence rate of under 0.1%, is not exempt from this global health challenge [1,3,4,5]. Over the past two decades, there has been a discernible downward trend in HIV transmission rates, which is a testament to the global strides in prevention and potential eradication [2].

Conversely, alarming spikes in HIV prevalence have been noted among the men who have sex with men (MSM) demographic, especially in specific Asian regions [5,6,7]. Across the continent, the sexual behavior of MSM has been thrust into the limelight, triggering societal concerns. This reaction is due to the escalating risk of HIV infections within this group. For example, a 2022 epidemiological report revealed that a staggering 99.1% of novel HIV infections were sexually transmitted. Notably, infections attributed to homosexual contact increased from 56.9% in 2018 to 60.3% in 2022 [4]. The intersection of sociocultural and epidemiological forces cast MSM in Korea as a particularly vulnerable group. However, the narrative surrounding MSM remains enigmatic in Korea, overshadowed by a plethora of studies on the general population. The societal discomfort associated with homosexuality and HIV/AIDS has stymied the growth of research on this nexus [1,5,6].

Such stigmatization erects formidable barriers to combating HIV/AIDS. Stigma and discrimination against MSM not only hinder research but also thwart national initiatives to amplify HIV cognizance. Entrenched misconceptions deter individuals from HIV testing or revealing their HIV-positive diagnosis and perpetuate myths about HIV propagation [1,5,6,8].

Such deeply ingrained stigmatization hinders the community’s receptiveness to health interventions and restricts research and national initiatives aimed at HIV education [9,10]. HIV-related stigma is a formidable deterrent to testing, prevention, and treatment. Systematically assessing the magnitude and implications of stigma is essential for devising counteractive strategies to enhance health outcomes [9,11].

It is paramount to recognize that societal perceptions and knowledge about HIV and the associated stigma can morph over time. Periodical assessments ensure adaptive adjustments to intervention and treatment programs, reflecting the prevailing societal pulse [5,12]. An intricate understanding of these misconceptions and societal dispositions is pivotal in shaping health campaigns, prevention techniques, and HIV-centric policies. Consistent research and community surveys yield invaluable insights, allowing policymakers and health organizations to customize their interventions adeptly [9]. Yet, a conspicuous research chasm exists regarding HIV prevention knowledge and attitudes amongst MSM in Korea [1]. Addressing this void is not merely an academic pursuit; it is pivotal to crafting efficacious HIV prevention programs and health strategies. The findings and insights are expected to inform and refine future interventions and policy directions.

## 2. Materials and Methods

### 2.1. Research Design and Study Population

This study incorporated a series of five cross-sectional surveys conducted over a decade—from 2012 to 2022—with the aim of assessing knowledge, attitudes, and practices concerning HIV among men who have sex with men (MSM) in Korea. The surveys were administered biennially, with one exception: no surveys were conducted in 2016. A self-administered questionnaire was employed for data collection in each survey, and different samples of participants were drawn for each iteration to ensure variability. The eligibility criteria for participants remained uniform across all surveys, targeting men aged 19–49 years who reported having had insertive or receptive anal intercourse with another man at least once. MSM is an inclusive public health construct that defines the sexual behaviors of men engaging in sexual activities with other men irrespective of their underlying motivations or identification with a particular community.

### 2.2. Participants and Procedures (2012–2022)

Between 2012 and 2022, a series of five cross-sectional surveys were conducted online to assess knowledge, attitudes, and behaviors related to HIV among MSM in Korea. A self-administered questionnaire was used for the surveys. 

Since 2012, IvanCity.com, a membership-based website, has served as the primary sampling platform for surveys. As a prominent platform for the MSM community in Korea, it has been instrumental in reaching the MSM population. In the current research, eligible members were notified of the survey via email or mobile alerts.

With the advent of social media and dating applications in 2018, our sampling strategy evolved. Recognizing the changing digital environment, we expanded our recruitment to include not only IvanCity but also popular dating apps, such as Jack’d and DIGSSO.

This study, which was conducted from 2012 to 2022, primarily used a questionnaire from the 2012 survey. Of the five surveys, the 2012, 2014, and 2018 surveys were approved by the Institutional Review Board of Seoul National University, and the 2020 and 2022 surveys were approved by the Institutional Review Board of Sahmyook University. In 2022, a new survey was approved by the Institutional Review Board of Sahmyook University along with a follow-up on the previous results (SYU 2022-07-007-002). The sample size was calculated to have approximately 384 participants to ensure representativeness and adequate statistical power based on an estimated 50% prevalence of correct responses, a 95% confidence level, and a 5% margin of error. However, each survey included more than 900 participants, far exceeding the minimum requirement. A larger sample size increased the statistical power of the study, ensuring that the results would be robust and reliably representative of the target MSM population in Korea. 

### 2.3. Measures

The survey questionnaire, originally developed in 2012, was primarily designed to assess HIV-related knowledge and stigma among MSM. To track changes in knowledge and attitudes, the original questions were largely retained in subsequent surveys. In 2018, the questionnaire was revised substantially to include five additional questions to enrich the assessment of HIV knowledge. This revision was in line with the indicators established by the United Nations General Assembly Special Session on HIV/AIDS (UNGASS) [13]. These enhancements were made while ensuring that the basic structure of the questionnaire remained intact, making it easier to compare data over time.

In 2018, questions regarding awareness, knowledge, and attitudes toward pre-exposure prophylaxis (PrEP) were included in the survey. However, while these data were collected, they were deemed beyond the scope of this particular study and were not included in the current analysis. By 2020, the tool was further refined with additional items specifically designed to assess HIV-related stigma, thus maintaining the continuity and integrity of the core objectives of the survey. 

#### 2.3.1. HIV-Related Knowledge

Significant enhancements were made from 2018 onwards in the survey methodology. While the initial grouping of five inquiries regarding knowledge related to HIV remained consistent, in line with surveys conducted before 2014 to ensure continuity and consistency over time, a more comprehensive analysis of the modes of HIV transmission was needed. In 2018, five new questions were added to focus on modes of transmission. The survey currently includes a comprehensive set of 10 questions, which has expanded the level of detail and depth of HIV-related knowledge data. The HIV/AIDS knowledge variable, with binary outcomes, was created by considering the responses to questions related to HIV/AIDS knowledge (Table 1). Responses categorized as “don’t know” were applied to answers where participants showed uncertainty or a lack of knowledge. Correct responses were coded as “1,” whereas incorrect responses were coded as “0.” The knowledge scores for questions 1–5 and questions 6–10 were summed. Additionally, the total knowledge scores for questions 1–10 were summed. The HIV knowledge score, comprising questions 1–5 across all surveys, was divided into high and low knowledge categories based on the opinions of HIV experts and the median scores. To differentiate between high and low levels of knowledge, a cutoff score was applied: 4 or higher scores were classified as high, while scores below 4 were categorized as low.

#### 2.3.2. HIV-Related Stigma

Stigma and discrimination frequently manifest as a type of social distancing toward specific groups, leading to the desire to separate individuals from regular social activities and interactions [7,8,10,14]. This inclination is particularly noteworthy in the context of HIV, where infected people may encounter opposition from others to share daily habits such as employment, cohabitation, and communal meals [6].

In the current study, HIV-related stigma was assessed using eleven questions: four items focused on common misconceptions about HIV, four focused on personal stigma against people living with HIV/AIDS (PWHA), and three focused on societal prejudice [15,16]. In 2020, three additional items were added to reflect increasing awareness of social stigma.

Participants were presented with statements such as “HIV is a dirty disease” and “If an employee is diagnosed with HIV, they ought to resign from their position.” It should be noted that the survey has evolved over time. In 2014, it consisted of seven questions. However, in recognition of the need for more comprehensive data, several other questions were added in 2018 and 2020, including “I would end a relationship if my partner revealed they were HIV positive” and “People should be fired for being HIV positive and working with others,” among others. 

HIV-related stigma was measured using 11 items that were responded to using a 5-point Likert scale. The eleven items formed three subscales: misconception of HIV/AIDS (four items), personal acceptance toward PWHA (four items), and societal stigma (three items). The three social stigma items were included in 2020 to evaluate broader societal stigma. Item responses ranged from 1 = “strongly disagree” to 5 = “strongly agree.” From 2012 to 2018, responses to seven items assessing the misconception of HIV/AIDS and personal acceptance toward PWHA were combined to form a total stigma score. In 2020 and 2022, the sum of 10 items formed the total score. Additionally, three items were reverse scored. Higher scores indicate greater HIV-related stigma. The HIV-related stigma score, which is the sum of questions 1 to 7 used in all surveys, was dichotomized into high and low stigma scores. A cutoff was used to define high stigma as a score of 16 (median) and above, and low stigma as a score below 16. The scale’s internal consistency reliability for the 10 items has historically been robust, with Cronbach’s alpha values between 0.80 in 2020 and 0.81 in 2022. The trend of HIV-related stigma for each item was determined by creating a binary response to each of the 10 items.

### 2.4. Data Analysis

Utilizing consistent items across the seven surveys helped examine trends in HIV-related knowledge and stigmatizing attitudes over the decade spanning from 2012 to 2022. For the dependent variables, responses to each item were coded as either 0 or 1. The percentages reported in Table 1 and Table 2 correspond to the responses coded as 1, which represent either a correct answer or a participant’s agreement with the statement.

Given the uneven intervals between survey years—specifically, a 2-year gap between the 2012 and 2014 surveys and a 4-year gap between the 2014 and 2018 surveys—an independent variable was created for the survey year. This variable was coded as follows: 0 for 2012, 2 for 2014, 6 for 2018, 8 for 2020, and 10 for 2022. The assumption was that the progression of knowledge and stigmatizing attitudes would be linear and consistent in the intervening years when no data were collected. Significant results, characterized by odds ratios with a *p*-value of less than 0.05, are highlighted in the results. Additionally, logistic regression was used to identify the independent variables (i.e., the demographic and social variables) that influence the dichotomous dependent variables, HIV-related knowledge, and stigma. All statistical analyses were performed using the R 4.3.2 software package.

## 3. Results

### 3.1. Sociodemographic Characteristics, Sexual Identity, and Coming out Status of the Participants (2012–2022)

The 19–29 age group showed a significant decrease in participation from 49.8% in 2012 to 26.6% in 2022. Conversely, the 40–49 age group showed an upward trend, peaking at 37.8% in 2022. While younger people were more likely to participate in the 2012 survey, all age groups were represented in the most recent survey, indicating a shift in the age distribution of the survey population. The majority of participants had a 4-year degree or higher. In particular, this group represented 58.7% of the sample in 2022. 

There was a significant increasing trend in participants who identified as gay, which rose to 82.9% in 2022. Simultaneously, there was a decline in participants who identified as bisexual, from 28.9% in 2012 to 15.6% in 2022. One encouraging observation was the gradual increase in the number of participants who voluntarily disclosed their sexual identity, reaching 30.8% by 2022. Simultaneously, the proportion of participants who had not come out decreased. However, a significant proportion of the participants had not come out and had no intention of doing so in the future. 

### 3.2. Knowledge Related to HIV/AIDS

Table 2 highlights the development of knowledge related to HIV/AIDS over the ten-year period. The response rates to the statements function as measures, revealing prevalent misconceptions and the extent of accurate knowledge regarding HIV/AIDS. The final column displays the outcomes of the statistical verification, which indicates the importance of the observed trends and patterns.

The first five statements maintained consistent levels of understanding across the five periodic surveys. The higher rate of correct responses to the statement regarding medical treatments for HIV/AIDS indicates a broad recognition of recent therapeutic progress, which has commendably persisted over the decade. Responses to the statements about sexual transmission demonstrated varying levels of knowledge. The proportion of correct responses to the question “Having sex with only one uninfected partner can reduce risk of HIV/AIDS transmission” decreased slightly from 67.9% in 2012 to 66.3% in 2022. Beliefs about transmission through mosquito bites (2012: 55.2%; 2022: 61.6%) and the sharing of drinking glasses (2012: 83.9%; 2022: 84.3%) remained similar. In 2022, approximately 40% of the participants held the erroneous belief that HIV/AIDS could be transmitted by mosquitoes, indicating an urgent need for education. The statement that even healthy-looking individuals might be infected with HIV achieved the highest correct answer rate of 94.9% in 2012 and 93% in 2022, emphasizing the widespread knowledge of the latent nature of the disease. The total mean score for items 1–5 was basically the same over the ten-year period: 3.9 in 2012, 3.9 in 2014, 3.8 in 2018, 3.8 in 2020, and 3.9 in 2022.

In 2018, five new items were included in the survey. Of these, the statement “unprotected sexual contact with an HIV-positive individual, even without ejaculation, can result in transmission” had the highest correct response rate at 84.0% in 2018. However, this proportion significantly decreased by 2022 (OR year = 0.80, *p* < 0.001). Another statement in 2018, “oral sex without ejaculation can still result in syphilis or gonorrhea infection,” had an accuracy rate of 78.9%. Moreover, 72.1% of the participants in 2018 acknowledged the increased risk for the receptive partner during unprotected anal intercourse with an HIV-positive individual. The total scores for items 6–10 were basically the same over the five-year period: 3.1 in 2018, 2.9 in 2020, and 3.0 in 2022.

However, the results indicate gaps in knowledge. In 2018, only 43.7% recognized that deep kissing does not pose a risk for HIV transmission, and this rate decreased to 39.4% by 2022. Another misconception held by 32.7% of the participants in 2018 and 31.7% in 2022 was that HIV has a higher transmission rate than other sexually transmitted infections (STIs) during intercourse with an HIV-positive person. This misunderstanding may exacerbate stigma and discrimination against individuals living with HIV. Furthermore, the correct response rates for items 6 and 7 declined from the 2012 levels in 2018 (OR year = 0.86, *p* < 0.001) and 2022 (OR year = 0.80, *p* < 0.001).

The multivariate logistic regression results demonstrated that HIV-related knowledge was significantly associated with age, education, and sexual identity in all five surveys, as outlined in Table 3. Surveys conducted in 2012 (OR = 1.83, *p* < 0.01), 2020 (OR = 2.51, *p* < 0.01), and 2022 (OR = 2.38, *p* < 0.01) revealed that individuals aged 40–49 consistently indicated higher levels of knowledge than their 19–29 age group counterparts, which was followed by the 30–39 age group. In 2012 (OR = 1.56, *p* < 0.05), 2014 (OR = 1.46, *p* < 0.05), 2020 (OR = 1.92, *p* < 0.01), and 2022 (OR = 1.46, *p* < 0.05), individuals aged 30–39 displayed greater levels of knowledge than those aged 19–29. Participants who completed at least a four-year college education consistently had higher levels of knowledge than those with a high school education or less (2012: OR = 2.01 *p* < 0.01; 2014: OR = 1.96, *p* < 0.01; 2018: OR = 1.92, *p* < 0.001; 2020: OR = 2.21, *p* < 0.05; 2022: OR = 1.44, *p* < 0.05). Surveys conducted in 2018 (OR = 0.21, *p* < 0.01) and 2022 (OR = 0.45, *p* < 0.05) revealed that married or separated individuals had considerably lower knowledge scores than unmarried individuals. Additionally, participants who identified as gay consistently exhibited higher levels of knowledge than those who identified as bisexual (2012: OR = 0.59 *p* < 0.01; 2014: OR = 0.51, *p* < 0.001; 2018: OR = 0.61, *p* < 0.01; 2020: OR = 0.64, *p* < 0.05; 2022: OR = 0.52, *p* < 0.001), transgender in 2012 (OR = 0.29, *p* < 0.01), and who indicated they were undecided in 2012 (OR = 0.44, *p* < 0.01) and 2018 (OR = 0.39, *p* < 0.01). Coming out status was not significantly related to HIV-related knowledge in any of the surveyed years.

### 3.3. HIV-Related Stigma

Table 4 outlines the percentage of participants who agreed with different statements regarding HIV-related stigma in three domains: misconceptions related to HIV/AIDS, personal acceptance toward PWHA, and societal stigma. The study measured the extent of HIV/AIDS misconceptions using four items and assessed acceptance of PWHA using another set of four items. In 2020, three additional items were included to assess broader societal stigma related to HIV.

The analysis of HIV/AIDS misconceptions indicated a marked trend: approximately one-third (33.3%) of the 2022 participants regarded “HIV as a filthy disease,” up from 25.4% in 2012, which was a significant increase (OR year = 1.04, *p* < 0.001). There was a significant decrease in the belief that an HIV-positive employee should resign from their job, from 12.8% in 2012 to 10.2% in 2022 (OR year= 0.98, *p* < 0.001). There was a peak in the belief that PWHA should be restricted to care facilities, with 13.2% in 2020 expressing such a view (OR year = 1.04, *p* < 0.05).

Positive attitudes toward PWHA, such as a willingness to share a meal with someone living with HIV, increased to 68.1% in 2022 from 60% in 2012, but the increase was not significant. Additionally, across the years, roughly three-quarters of the participants (from 70.4% to 83.9%) were open to living in the same household with an HIV-infected family member or the same neighborhood as someone with HIV, but this significantly decreased by 2020 (family: OR year = 0.96, *p* < 0.05; neighbor: OR year = 0.97, *p* < 0.05).

The last three items demonstrate prevailing social stigma. By 2022, 58.1% of the participants believed that if they had to work with someone with HIV, most people would want to fire them. A total of 77.7% believed that most people would be reluctant to have dinner with someone with HIV, and 73.2% believed that most people would not associate with someone with HIV in their neighborhood. The total stigma score for items 1–7 was basically the same over the ten-year period: 16.4 in 2012, 16.2 in 2014, 16.4 in 2018, 16.8 in 2020, and 15.9 in 2022. The total stigma score for items 1–10 was 27.8 in 2020 and 16.4 in 2022.

Multivariate logistic regression analysis revealed significant associations between HIV-related stigma and certain sociodemographic characteristics (Table 5). Age was a prominent determinant of stigma in 2018 and 2020. In 2012 (OR = 0.59, *p* < 0.001) and 2014 (OR = 0.69, *p* < 0.05), participants who completed at least a four-year college education had a significantly lower level of stigma than those with a high school education or less. Marital status was significantly associated with stigma in 2012 (OR = 2.27, *p* < 0.01) and 2022 (OR = 2.70, *p* < 0.01). Additionally, coming out status was a significant factor influencing stigma perceptions in all surveys (*p* < 0.05). Older participants and those with higher levels of education typically reported lower levels of HIV-related stigma. Conversely, higher levels of stigma were more prevalent among married individuals, those who identified as bisexual, and those who either had not disclosed their sexual identity or had no intention of doing so.

## 4. Discussion

This decade-long study provides a comprehensive snapshot of the evolution and involution of HIV-related knowledge and stigma, tracing the complex interplay of societal attitudes and awareness over time. It supports the hypothesis that MSM have a better understanding of HIV than the general population [16,17,18]. However, this understanding appears to have stagnated in certain aspects, if not declined. A decline in the correct answer rate to questions introduced in 2018, particularly regarding the mode of HIV transmission, highlights persistent misconceptions. It is noteworthy that knowledge did not decline in 2022, despite the interruption of HIV prevention programs in 2020 due to the COVID-19 pandemic [4].

Although the participants demonstrated a remarkable understanding of the nuances of HIV transmission, the persistence of misconceptions underscores the urgent need for sustained educational efforts. These misconceptions not only hinder prevention but also exacerbate the stigma experienced by people living with HIV. An understanding of HIV is critical not only for prevention but also for fostering a more inclusive society [6,19]. 

The fluctuating dynamics of sexual identity over the decade provide additional insights. The increase in the number of men who identified as bisexual may indicate an elevated level of acceptance of their sexuality, suggesting broader societal changes in the comprehension and acceptance of diverse sexual orientations. Nonetheless, a more thorough and qualitative investigation is necessary to ascertain the magnitude of this shift. The coming-out dynamics observed in this study showed positive development. The rise in voluntary disclosures, coupled with the decline in those who had not come out, could be indicative of reduced social stigma, reflecting the changing societal attitudes or more supportive community environments [1]. This trend is consistent with the findings of studies that noted that reduced stigma and greater social acceptance have empowered more people within the MSM community to openly acknowledge and discuss their sexuality [5,6,7].

Expanding on the issue of stigma, three distinct dimensions were identified and classified: misconceptions of HIV/AIDS, personal acceptance toward PWHA, and societal stigma [16]. There were reservations regarding misunderstandings pertaining to HIV/AIDS, as two out of four inquiries illustrated a statistically significant variation. People still perceive HIV/AIDS as an embarrassing sexually transmitted disease and assume accountability for those influenced. Nevertheless, favorable modifications in personal endorsement were noticed. The inclination to partake in a meal with an individual with HIV has grown from 60% in 2012 to 68.1% in 2022. Similarly, approximately 75% of respondents expressed their acceptance of cohabitating with family and neighbors with HIV, which echoes the results of Kim and Yang’s (2021) study [16]. This willingness is attributed to a better comprehension that HIV/AIDS cannot be transmitted through routine activities. It is noteworthy that subjective judgments have been omitted from this report. These findings correspond with surveys conducted among the general public and MSM. The social stigma associated with HIV decreased in the 2022 survey compared to 2020. However, additional research is necessary to accurately identify these trends, as items designed specifically to measure societal stigma were only introduced in 2020.

In Korean society, especially among older individuals, historically, significant stigma has been associated with homosexual identification. This societal pressure often leads individuals, especially those in opposite-sex marriages, to refrain from self-identifying as “gay” or “bi”, even if they are in same-sex relationships. This phenomenon is reflected in our data, in which a higher percentage of participants identified as “gay” in the most recent survey than in previous surveys (83% in the most recent survey versus 64.7% in previous surveys). This shift may reflect a change in social attitudes and acceptance of homosexual identities among younger generations in Korea rather than a true shift in social identity phenomena.

In addition, the study’s findings suggest that older age groups are more likely to have personal relationships with someone affected by HIV compared to younger participants. This observation may provide additional context for understanding attitudes toward HIV among different age groups in Korea. This suggests a generational difference in exposure to and response to the HIV/AIDS epidemic, which may influence the degree of internalization of HIV-related messages. Thus, it is crucial to interpret these findings with a keen understanding of the evolving social and linguistic landscape in Korea, where acceptance and self-identification of sexual minorities have been rapidly changing, especially among younger generations.

It is imperative that an eclectic coalition of non-governmental organizations lead efforts to create an inclusive atmosphere that recognizes and embraces individuals living with HIV/AIDS. Despite significant advancements in personal acceptance toward PWHA and increasing willingness to engage in daily activities with HIV-positive individuals, misconceptions about HIV/AIDS continue to exist. It is worrisome that HIV/AIDS is still primarily perceived through a lens of shame and culpability by many individuals. The need to eliminate stigma toward PWHA within the MSM community in Korea is of great importance. Such efforts not only uphold the principles of human rights and social justice for PWHA but also combat prevailing feelings of fear and shame, which may deter individuals from seeking medical care [1,9]. Such hesitations may worsen health issues and unintentionally spread the virus. By reducing stigma, engagement with HIV-focused services ranging from prevention to treatment can be strengthened. This is essential for curbing the spread of the virus and ensuring improved health outcomes for those affected [5]. The observed trends in societal stigma, although based on a limited dataset, are promising. However, further longitudinal studies are required for validation. 

This study spanned five waves of surveys over a decade and followed a consistent methodology across each iteration Although the methodology remained unchanged, the variability in the survey population between years could have affected the robustness of the longitudinal analysis. This limitation is critically considered in our discussion, and we emphasize that caution should be exercised when interpreting trends or changes over time. 

## 5. Conclusions

In conclusion, this study presents an overview of HIV-related knowledge and stigma, highlighting the need for ongoing, personalized interventions and the unwavering diligence of the academic and medical sectors. Ongoing assessments of HIV-related knowledge and stigma not only aid in determining the current level of awareness and prejudices in communities but also play a critical role in shaping interventions, policies, and strategies that address the changing dynamics of the HIV/AIDS epidemic. Active policy interventions, including counseling, testing, education, promotion, and the development of preventive programs, are crucial for promoting an inclusive environment. Reducing stigma and promoting the acceptance of PWHA not only ensures easy access to prevention, testing, and treatment services but also reduces the negative social and economic consequences faced by this community. Furthermore, these endeavors align with the wider goal of maintaining human rights and advocating for social justice. Moving forward, these insights should influence joint efforts to create an understanding society.

## Figures and Tables

**Table 1 healthcare-11-03135-t001:** Sociodemographic characteristics, sexual identity, and coming out status of MSM respondents (2012–2022).

Variable	Survey Year	2012 (N = 947)		2014 (N = 873)		2018 (N = 1002)		2020 (N = 1093)		2022 (N = 1085)	
		n	%	n	%	n	%	n	%	n	%
Age	19–29	472	49.8	431	49.4	511	51.0	407	37.2	289	26.6
	30–39	280	29.6	261	29.9	313	31.2	396	36.2	386	35.6
	40–49	195	20.6	181	20.7	178	17.8	290	26.5	410	37.8
Education	High school or less	306	32.3	263	30.1	328	32.7	345	31.6	263	24.2
	2 yrs of college	139	14.7	146	16.7	161	16.1	192	17.6	185	17.1
	4 yrs of college or more	502	53.0	464	53.2	513	51.2	556	50.9	637	58.7
Marital status (Heterosexual)	Single (Never married)	862	91.0	807	92.4	989	98.7	1076	98.4	1033	95.2
	Married/Separated/Widowed	85	9.0	66	7.6	13	1.3	17	1.6	52	4.8
Sexual identity	Gay	613	64.7	576	66.0	760	75.8	896	82.0	899	82.9
	Transgender	23	2.4	13	1.5	6	0.6	15	1.4	13	1.2
	Bisexual	274	28.9	256	29.3	213	21.3	159	14.5	169	15.6
	Undecided	37	3.9	28	3.2	23	2.3	23	2.1	4	0.4
Coming out	Voluntarily	162	17.1	157	18.0	202	20.2	331	30.3	334	30.8
	Involuntarily outed	32	3.4	25	2.9	36	3.6	62	5.7	45	4.1
	Have not come out	148	15.6	130	14.9	187	18.7	151	13.8	173	15.9
	Have not come out, do not indent to	605	63.9	561	64.3	577	57.6	549	50.2	533	49.1
Total		947	100.0	873	100.0	1002	100.0	1093	100.0	1085	100.0

**Table 2 healthcare-11-03135-t002:** Percentage responding with the “Correct Answer” to HIV-related knowledge items.

Item	2012 (N = 947)	2014 (N = 873)	2018 (N = 1002)	2020 (N = 1093)	2022 (N = 1085)	OR Year ^a^ (95% CI)
1. With proper treatment, people with HIV/AIDS can live healthily (T)	83.3	84.0	81.0	79.5	82.2	NS
2. Having sex with only one uninfected partner can reduce the risk of HIV/AIDS transmission (T)	67.9	69.0	67.3	65.7	66.3	NS
3. Getting bitten by a mosquito that previously bit someone with HIV/AIDS can lead to infection (F)	55.2	56.7	53.6	59.6	61.6	NS
4. Sharing a drinking glass with someone with HIV/AIDS can lead to infection (F)	83.9	84.4	83.9	82.3	84.3	NS
5. Even healthy-looking people can be infected with HIV (T)	94.9	95.2	95.1	93.4	93.0	NS
Total score for questions 1 to 5: Mean (SD)	3.9 (1.2)	3.9 (1.1)	3.8 (1.2)	3.8 (1.3)	3.9 (1.1)	-
6. Deep kissing cannot transmit HIV/AIDS (T)	NA	NA	43.7	36.7	39.4	0.86 *** (0.79–0.94)
7. Unprotected sexual contact, even in the absence of ejaculation, with an HIV-positive individual can result in transmission (T)	NA	NA	84.0	76.9	75.8	0.80 *** (0.71–0.89)
8. During unprotected anal sex with a person with HIV/AIDS, being in the receptive (bottom) position increases infection risk (T)	NA	NA	72.1	71.4	75.1	NS
9. Oral sex without ejaculation can still lead to syphilis or gonorrhea infection (T)	NA	NA	78.9	76.0	78.0	NS
10. The probability of HIV/AIDS transmission during sex with an infected person is lower than the probability of transmission for other STIs (T)	NA	NA	32.7	29.8	31.7	NS
Total score for questions 6 to 10: Mean (SD)	-	-	3.1 (1.2)	2.9 (1.3)	3.0 (1.3)	-
Total score for questions 1 to 10: Mean (SD)	-	-	6.9 (2.0)	6.7 (2.2)	6.9 (2.1)	-

T = true, F = false, OR = odds ratio; NA = item not asked that year; NS = not significant; STIs = sexually transmitted infections; ^a^ Figures in the “OR Year” column are odds ratios, indicating the annual change in the likelihood of endorsing the item. *** *p* < 0.001.

**Table 3 healthcare-11-03135-t003:** Logistic regression analysis to determine factors of HIV-related knowledge.

Variable	2012	2014	2018	2020	2022
	OR (95% CI)	OR (95% CI)	OR (95% CI)	OR (95% CI)	OR (95% CI)
Age					
19–29	ref.	ref.	ref.	ref.	ref.
30–39	1.56 * (1.09, 2.24)	1.46 * (1.02, 2.09)	1.05 (0.76, 1.44)	1.92 ** (1.39, 2.64)	1.46 * (1.04, 2.04)
40–49	1.83 ** (1.17, 2.87)	1.34 (0.86, 2.09)	1.60 (1.06, 2.42)	2.51 ** (1.74, 3.61)	2.38 ** (1.66, 3.41)
Education					
High school or less	ref.	ref.	ref.	ref.	ref.
2 yrs of college	0.93 (0.60, 1.42)	1.41 (0.90, 2.19)	1.37 (0.91, 2.06)	1.00 (0.68, 1.47)	1.03 (0.67, 1.56)
4 yrs of college or more	2.01 ** (1.44, 2.80)	1.96 ** (1.40, 2.75)	1.92 *** (1.40, 2.62)	2.21 ** (1.62, 3.01)	1.44* (1.04, 1.99)
Marital status					
Single					
Married/Separated/Widowed	1.05 (0.57, 1.91)	0.69 (0.38, 1.27)	0.21 ** (0.07, 0.69)	3.10 (0.68, 14.16)	0.45 * (0.25, 0.84)
Sexual identity					
Gay	ref.	ref.	ref.	ref.	ref.
Transgender	0.29 ** (0.12, 0.70)	0.39 (0.13, 1.23)	0.20 (0.03, 1.14)	0.73 (0.24, 2.17)	0.91 (0.27, 3.10)
Bisexual	0.59 ** (0.42, 0.82)	0.51 *** (0.37, 0.72)	0.61 ** (0.43, 0.85)	0.64 * (0.44, 0.93)	0.52 *** (0.37, 0.75)
Undecided	0.44 ** (0.22, 0.89)	0.52 (0.23, 1.16)	0.39 ** (0.17, 0.91)	0.69 (0.29, 1.68)	1.44 (0.14, 14.56)
Coming out					
Voluntarily	ref.	ref.	ref.	ref.	ref.
Outing	0.48 (0.21, 1.07)	1.37 (0.53, 3.55)	0.58 (0.28, 1.22)	0.63 (0.35, 1.13)	0.65 (0.33, 1.27)
Have not come out	0.68 (0.42, 1.11)	1.33 (0.79, 2.23)	0.67 (0.43, 1.04)	0.72 (0.47, 1.10)	0.81 (0.54, 1.22)
Have not come out, do not indent to	1.00 (0.67, 1.52)	1.21 (0.81, 1.82)	0.83 (0.57, 1.21)	0.77 (0.56, 1.06)	0.88 (0.64, 1.22)

Ref. = reference group; OR = odds ratio. The HIV knowledge score, which comprises questions 1–5 across all surveys, was divided into high and low knowledge. Scores of 4 or higher were classified as high, while scores below 4 were classified as low. * *p* < 0.05, ** *p* < 0.01, *** *p* < 0.001.

**Table 4 healthcare-11-03135-t004:** Percentage responding with “Agree” or “Strongly Agree” to HIV-related stigma items.

Item	2012 (N = 947)	2014 (N = 873)	2018 (N = 1002)	2020 (N = 1093)	2022 (N = 1085)	OR Year ^a^
Misconceptions of HIV/AIDS						
1. HIV is a filthy disease	25.4	25.8	28.4	31.9	33.3	1.04 *** (1.02–1.06)
2. If an employee is infected with HIV, they should resign from their job	12.8	11.5	9.9	11.8	10.2	0.98 *** (0.95–1.01)
3. If someone becomes infected with HIV, it is their own responsibility	41.6	41.7	38.8	45.4	43.5	NS
4. PWHA should be isolated and sent to care facilities	10.6	10.3	12.0	13.2	10.1	1.04 * (1.00–1.07)
Personal acceptance of PWHA						
5. I can share a meal with someone infected with HIV ^†^	60.0	61.6	58.9	61.9	68.1	**NS**
6. If someone in my family becomes infected with HIV, I can live together at home ^†^	76.7	77.0	72.1	70.4	75.8	0.96 * (0.93–0.98)
7. I can live in the same neighborhood as someone infected with HIV ^†^	77.4	78.1	73.4	74.3	83.9	0.97 * (0.95–0.99)
Total stigma score for questions 1 to 7: Mean (SD)	16.4 (5.4)	16.2 (5.1)	16.4 (5.5)	16.8 (5.8)	15.9 (5.2)	-
Societal stigma						
8. If they had to work with someone with HIV, most people would want to fire them	NA	NA	NA	64.9	58.1	-
9. Most people would be reluctant to have dinner with someone with HIV	NA	NA	NA	80.2	77.7	-
10. Most people would not associate with someone with HIV in their neighborhood	NA	NA	NA	77.0	73.2	-
Total stigma score for questions 1 to 10: Mean (SD)	-	-	-	27.8 (5.6)	26.4 (5.3)	

NA = item not asked that year; NS = not significant; PWHA = people living with HIV/AIDS. Indicates that no statistical tests were performed. ^a^ Figures in the “OR Year” column represent odds ratios, indicating the annual change in the probability of endorsing the respective item. * *p* < 0.05, *** *p* < 0.001. ^†^ When participants responded with “Agree” or “Strongly agree” to the item, it suggests a low level of stigma associated with the topic. Items 5, 6, and 7 were reverse scored.

**Table 5 healthcare-11-03135-t005:** Logistic regression analysis to determine factors of HIV-related stigma.

	2012	2014	2018	2020	2022
	OR (95% CI)	OR (95% CI)	OR (95% CI)	OR (95% CI)	OR (95% CI)
Age					
19–29	ref.	ref.	ref.	ref.	ref.
30–39	0.80 (0.58, 1.11)	0.76 (0.55, 1.06)	0.70 * (0.51, 0.94)	0.73 * (0.53, 0.99)	0.85 (0.61, 1.17)
40–49	0.71 (0.48, 1.06)	0.83 (0.55, 1.24)	0.64 * (0.45, 0.92)	0.87 (0.62, 1.23)	0.88 (0.63, 1.22)
Education					
High school or less	ref.	ref.	ref.	ref.	ref.
2 yrs of college	0.84 (0.55, 1.28)	1.19 (0.77, 1.85)	1.11 (0.75, 1.65)	1.39 (0.94, 2.06)	1.21 (0.81, 1.81)
4 yrs of college or more	0.59 *** (0.43, 0.81)	0.69 * (0.50, 0.96)	1.21 (0.89, 1.63)	0.78 (0.58, 1.05)	0.83 (0.61, 1.13)
Marital status					
Single					
Married/Separated/Widowed	2.27 ** (1.29, 3.99)	1.28 (0.69, 2.37)	1.84 (0.55, 6.13)	1.00 (0.35, 2.84)	2.70 ** (1.35, 5.39)
Sexual identity					
Gay	ref.	ref.	ref.	ref.	ref.
Transgender	1.82 (0.74, 4.48)	2.13 (0.63, 7.27)	4.09 (0.47, 35.91)	4.35 (0.96, 19.66)	10.87 * (1.38, 85.83)
Bisexual	1.70 *** (1.24, 2.33)	2.01 *** (1.45, 2.80)	1.40 * (1.01, 1.95)	1.38 (0.95, 2.02)	2.14 *** (1.49, 3.09)
Undecided	2.99 ** (1.40, 6.39)	2.86 * (1.17, 6.96)	1.10 (0.47, 2.57)	0.62 (0.27, 1.45)	0.88 (0.12, 6.41)
Coming out					
Voluntarily	ref.	ref.	ref.	ref.	ref.
Outing	2.44 * (1.08, 5.50)	0.97 (0.41, 2.32)	2.35 * (1.14, 4.85)	2.58 * (1.44, 4.62)	1.67 (0.88, 3.18)
Have not come out	1.35 (0.85, 2.14)	1.05 (0.65, 1.69)	1.79 ** (1.19, 2.70)	1.42 (0.96, 2.11)	1.47 * (1.01, 2.15)
Have not come out, do not indent to	1.57 * (1.08, 2.29)	1.85 *** (1.27, 2.71)	2.49 *** (1.76, 3.52)	2.35 *** (1.75, 3.16)	1.91 *** (1.42, 2.58)

Ref. = reference group; OR = odds ratio. The HIV-related stigma score, which is the sum of items 1 to 7 used in all surveys, is dichotomized into high and low stigma. High stigma was a score of 16 and above, and low stigma was a score below 16. * *p* < 0.05, ** *p* < 0.01, *** *p* < 0.001.

## Data Availability

Any queries regarding the data used in this study may be directed to the corresponding author. The dataset used in the present study is available upon reasonable request.

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
