# Peer review of "Trends in Human Immunodeficiency Virus-Related Knowledge and Stigma among Men Who Have Sex with Men in the Republic of Korea from 2012 to 2022"

_healthcare, 2023, doi:10.3390/healthcare11243135_

Round 1
Reviewer 1 Report
Comments and Suggestions for Authors
I have thoroughly evaluated the manuscript titled "Trends in HIV-Related Knowledge and Stigma Among Men Who Have Sex With Men in the Republic of Korea from 2012 to 2022," submitted to the journal HEALTHCARE. Undoubtedly, the study presents interesting data that could contribute to guiding public policies. However, several points lead me to not recommend the publication of this manuscript, which I detail below:
1. Changes in the questionnaire over the years: It is not clear what changes were made to the questionnaire over the studied period. The text only mentions that changes were made but does not provide a clear understanding of the impact of these alterations over time. A relevant example is the evolution of understanding regarding unprotected sex, which in 2012 was considered unprotected for HIV/AIDS, but in 2018, with the availability of PrEP, perceptions may have changed. The lack of analysis on how these changes affect the results is a significant gap.
2. Missing sample size calculation: The manuscript lacks any sample size calculation that demonstrates the representativeness of the results for each year and for the entire study period. The absence of this essential information undermines the validity of the results and their broader applicability.
3. Missing intra-group analyses: For a study of this kind, intra-group analyses are necessary to provide a deeper understanding of trends and differences over time. The absence of these analyses limits the usefulness of the study in identifying patterns and underlying factors.
4. Unaddressed limitations and biases: The manuscript presents numerous limitations and potential biases that are neither identified nor discussed by the author. It is crucial for the author to acknowledge and discuss these limitations so that readers can assess the robustness of the results more comprehensively and critically.
Based on these considerations, my recommendation is the rejection of the manuscript in its current form.
Author Response
I would like to express my gratitude to all the careful reviewers who have examined my manuscript. Thanks to the previous reviews, I was able to identify clear flaws in my manuscript and have worked hard to write a better paper.
Reviewer 1, Comment 1
Changes in the questionnaire over the years: It is not clear what changes were made to the questionnaire over the studied period. The text only mentions that changes were made but does not provide a clear understanding of the impact of these alterations over time. A relevant example is the evolution of understanding regarding unprotected sex, which in 2012 was considered unprotected for HIV/AIDS, but in 2018, with the availability of PrEP, perceptions may have changed. The lack of analysis on how these changes affect the results is a significant gap.
Response: I appreciate the valuable feedback. Questionnaires were used for HIV knowledge and attitudes in each of the five surveys; I described the survey process in more detail in the paper, and for PrEP we conducted a survey in 2018, which I do not describe as a finding in the paper but supplement in the discussion. I've colored the modified parts in red. Additionally, the following description has been included:
2.1 Research Design and Study Population.
This study incorporated a series of five cross-sectional surveys conducted over a decade—from 2012 to 2022—with the aim of assessing knowledge, attitudes, and practices concerning HIV among men who have sex with men (MSM) in Korea. The surveys were administered biennially, with one exception: no surveys were conducted in 2016. A self-administered questionnaire was employed for data collection in each survey, and different samples of participants were drawn for each iteration to ensure variability. The eligibility criteria for participants remained uniform across all surveys, targeting men aged 19-49 years who reported having had insertive or receptive anal intercourse with another man at least once. MSM is an inclusive public health construct that defines the sexual behaviors of men engaging in sexual activities with other men irrespective of their underlying motivations or identification with a particular community.
2.2 Participants and Procedures (2012–2022)
2.1 Research Design
Between 2012 and 2022, a series of five cross-sectional surveys were conducted online to assess knowledge, attitudes, and behaviors related to HIV among MSM in Korea. A self-administered questionnaire was used for the surveys.
2.2 Participants and Procedures (2012–2022)
Since 2012, IvanCity.com, a membership-based website, has served as the primary sampling platform for surveys. As a prominent platform for the MSM community in Korea, it has been instrumental in reaching the MSM population. In the current research, eligible members were notified of the survey via email or mobile alerts.
With the advent of social media and dating applications in 2018, our sampling strategy evolved. Recognizing the changing digital environment, we expanded our recruitment to include not only IvanCity but also popular dating apps such as Jack'd and DIGSSO.
This study, which was conducted from 2012 to 2022, primarily used a questionnaire from the 2012 survey. Of the five surveys, the 2012, 2014, and 2018 surveys were approved by the Institutional Review Board of Seoul National University, and the 2020 and 2022 surveys were approved by the Institutional Review Board of Sahmyook University. In 2022, a new survey was approved by the Institutional Review Board of Sahmyook University along with a follow-up on the previous results (SYU 2022-07-007-002). The sample size was calculated to have approximately 384 participants to ensure representativeness and adequate statistical power based on an estimated 50% prevalence of correct responses, a 95% confidence level, and a 5% margin of error. However, each survey included more than 900 participants, far exceeding the minimum requirement. A larger sample size increased the statistical power of the study, ensuring that the results would be robust and reliably representative of the target MSM population in Korea.
2.3. Measures
The survey questionnaire, originally developed in 2012, was primarily designed to assess HIV-related knowledge and stigma among MSM. To track changes in knowledge and attitudes, the original questions were largely retained in subsequent surveys. In 2018, the questionnaire was revised substantially to include five additional questions to enrich the assessment of HIV knowledge. This revision was in line with the indicators established by the United Nations General Assembly Special Session on HIV/AIDS (UNGASS) [13]. These enhancements were made while ensuring that the basic structure of the questionnaire remained intact, making it easier to compare data over time.
In 2018, questions regarding awareness, knowledge, and attitudes toward pre-exposure prophylaxis (PrEP) were included in the survey. However, while these data were collected, they were deemed beyond the scope of this particular study and were not included in the current analysis. By 2020, the tool was further refined with additional items specifically designed to assess HIV-related stigma, thus maintaining the continuity and integrity of the core objectives of the survey.
Reviewer 1, Comment 2
Missing sample size calculation: The manuscript lacks any sample size calculation that demonstrates the representativeness of the results for each year and for the entire study period. The absence of this essential information undermines the validity of the results and their broader applicability.
Response: I appreciate the valuable feedback. I elaborated on the description of the sample size in the revised article as follows.
The sample size was calculated to have approximately 384 participants to ensure representativeness and adequate statistical power based on an estimated 50% prevalence of correct responses, a 95% confidence level, and a 5% margin of error. However, each survey included more than 900 participants, far exceeding the minimum requirement. A larger sample size increased the statistical power of the study, ensuring that the results would be robust and reliably representative of the target MSM population in Korea.
Reviewer 1, Comment 3
Missing intra-group analyses: For a study of this kind, intra-group analyses are necessary to provide a deeper understanding of trends and differences over time. The absence of these analyses limits the usefulness of the study in identifying patterns and underlying factors.
- The reviewer's point is a good one, but since this paper analyzed five surveys over a 10-year period, it is difficult to describe the within-group analysis in detail due to the limited space of the paper. This is something we hope to do in future studies.
Reviewer 1, Comment 4
Unaddressed limitations and biases: The manuscript presents numerous limitations and potential biases that are neither identified nor discussed by the author. It is crucial for the author to acknowledge and discuss these limitations so that readers can assess the robustness of the results more comprehensively and critically.
Response: I appreciate the valuable feedback. This paper was conducted over a five-year period, and although the methodology was the same, the same people were not surveyed, so the results may be biased. We have described this as a limitation in the discussion. I discussed it as a limitation as follows.
This study spanned five waves of surveys over a decade and followed a consistent methodology across each iteration Although the methodology remained unchanged, the variability in the survey population between years could have affected the robustness of the longitudinal analysis. This limitation is critically considered in our discussion, and we emphasize that caution should be exercised when interpreting trends or changes over time.

Reviewer 2 Report
Comments and Suggestions for Authors
Thank you for the opportunity to review ‘Trends in HIV-related knowledge and stigma among men who have sex with men in the Republic of Korea from 2012 to 2022’. This study reports the outcomes of five cross sectional KABB studies carried out with MSM over a ten-year period with sample sizes ranging from 873 to 1093. I am deeply sympathetic to the ambitions of the author in this paper, and my recommendations come with a view to strengthening what could be a valuable paper.
There is no literature review per se, but the introduction provides context and background to the HIV situation in Korea. Once the author has considered my comments below, they may wish to develop the introduction a little further.
Methodology: The description of the evolution (and rationale) of the surveys is very well explained, and I commend the author for that clarity. The binary approach to responses in an East Asian context is appropriate, although it would be useful to hear how the author addressed the usual responses that tend toward the centre on the Likert scaled items (line 133). The Cronbach’s alphas are very respectable and consistent. However, information about participant recruitment is quite sparse: it appears that the surveys were launched on various social media platforms used by MSM but were largely passive in respect of recruiting respondents. It is not clear how long the studies were on each platform, how self-selecting the participants were, or when the author decided that they had sufficient responses to close the surveys. What was the sampling frame compared to the number of participants? It is not clear that the surveys were limited to Korean residents only; were Koreans living in other countries recruited or able to access these surveys? Were there checks in place to prevent multiple submissions by a single participant? We need to know more about the recruitment and participation strategy.
Information about ethics is also quite limited. In Line 78 the author refers to ‘this study’, but it is not clear whether the IRB approval is referring only to the meta-analysis discussed in the present paper, or some other study, and particularly the 2012 study on which the present paper claims to rely most. If the ethics review relates only to the present meta-analysis, then it would be important to describe what review/approval process was undertaken in in the five individual precursor studies. Ethics reviews and approvals cannot be carried out retrospectively. It would appear that the 2012 (IvanCity.com) study was not anonymous, since participants were recruited by email or mobile phone; to what extent was anonymity assured in the five individual studies, and particularly in the 2012 study? Being more explicit about the ethics of the five studies is an essential addition to this paper.
Results: The author speaks of age groups as though they were discrete cohorts (lines 167-179). What is happening here, I think, is that the later studies confirm whether or not a particular age group is more or less likely to use an app, participate in a study, or to identify with a particular identity (like gay). It would also be helpful if the author could identify what Korean words were used in the identity categories: gay, trans, bi, etc. are Western identity labels with which non-English speakers (or ‘non-Westerners’) may not identify, at least in the early iterations of the study. Opposite-sex married Korean men are (or were) unlikely to identify themselves as gay or bi even when they had sex with men; they are (or were) also less likely to internalise HIV-safer sex messages, relying on the hetero-marriage to protect them. The data would seem to bear this out, as in the most recent study nearly 83% of participants identified as ‘gay’, but in the earlier one only 64.7% did. This could easily be explained as an artifact of language rather than a Korean social identity phenomenon. It may be worth pointing out (lines 226-229) that the 40-49 age groups in the various studies were more likely to have know someone with HIV or AIDS personally than the younger age groups; this may also explain the finding reported in lines 287-88.
It is also worth noting in the discussion about Table 4 that items 8-10 are about attributed attitudes—what participants think other people will think—so I’m not quite sure what these items are measuring. Attribution has its own theoretical literature, so these items should be reported and interpreted with caution. It may be useful to consult some of the attribution theory here.
Discussion: The author should claim that this is a ‘decade-long’ study only with the greatest caution. The author does not suggest anywhere else that these five studies were conceived of as a single study (nor does there appear to be an ethics review of the study series as a whole before the 2012 survey), and the different instruments often resulted in missing or non-comparable data on several items. While certainly instruments can evolve, I don’t think this sequence of surveys can be characterised as a single study. It is more of a meta-analysis. That said, it is more than a ‘snapshot’ (line 297), a word usually used to describe a single cross-sectional study. This is certainly a longitudinal analysis which reports trends in data over time. The discussion in lines 311-322 is very interesting, but in my view avoids what may be two salient social phenomena: one is that the older age groups are more likely to have known someone with HIV, either in Korea or internationally (or certainly been exposed to heightened media attention in the late 1990s; the other is the rise in the K-pop phenomenon which models the fluidity of male sexuality, homosociality, and homoeroticism in a very public way. I think the data support the author’s broad conclusions set out in lines 344-361.
There are 19 references, of which 5 (26.3%) were published in the most recent 5 years (2019-2023), an additional 5 (26.5%) in the most recent 10 years, and 2 (10.5%) more in the most recent 15 years (2009-2023). This means that 35.8% of the references are older than 15 years, which is quite high for a paper about HIV; only about half were published in the most recent 10 years. A lot happens in HIV in 10 years. Given the sparseness of the literature cited, the author will want to consider carefully whether the recent literature has anything to contribute to their analysis, given that they are presenting it as a new paper in 2023. It seems to me that the UNAIDS literature on stigma, and the large literature on the impact of stigma would strengthen the context for this study. Certainly the study is data-heavy, but that does not mean that the data by itself tells the story. The author makes claims that reflect values, e.g., ‘One encouraging observation’ (line 175), but does not support these value claims with literature (why is this observation encouraging?). The consequences of stigma are apparent in the literature, but that literature is not cited. Citing the UNAIDS goals of ‘Getting to Zero’, for instance, would be a low-effort way of creating a theoretical and public health rationale for this paper. Becoming more familiar with the recent theoretical literature on HIV would strengthen the paper considerably.
Minor issues
Line 35: ‘cast’ is not quite the right word; consider ‘sets’ or ‘establishes’.
Line 36: ‘overshadowed by a plethora of studies on the general population’: this is unclear. Does this refer to a plethora of HIV-related studies? Were these epidemiological studies, KABB studies, or something else?
Line 108: ‘was’ should be ‘were’.
Table 1: A very minor issue, but it would make the table much easier to read if heavier vertical lines separated the various studies from each other; once the reader is in the lower rows of the table it is difficult to track which year’s data is being reported, particularly as it breaks across pages. This is not so much an issue in Table 2, where there is only one column of data per year.
In Tables 1 and 3 I’m not sure what the word ‘Outing’ means- does this mean ‘Outed’ by someone else? Perhaps ‘Voluntary disclosure’ and ‘Involuntary outed’ would be clearer.
Line 199: ‘HIV/AIDS carriers’ is stigmatising language in English and should be avoided (UNAIDS. (2015). UNAIDS terminology guidelines. http://www.unaids.org/sites/default/files/media_asset/2015_terminology_guidelines_en.pdf )

I have made a couple of very minor recommendations related to clarity of English in my narrative. The English is of a very high standard otherwise.
Author Response
I would like to express my gratitude to all the careful reviewers who have examined my manuscript. Thanks to the previous reviews, I was able to identify clear flaws in my manuscript and have worked hard to write a better paper.

Reviewer 3 Report
Comments and Suggestions for Authors
The article titled "Trends in HIV-Related Knowledge and Stigma Among Men Who Have Sex with Men in the Republic of Korea from 2012 to 2022" has several significant shortcomings, which are outlined below: The clarity of the method is lacking. A series of surveys were done over a ten-year period, from 2012 to 2022. However, it is worth noting that the process for determining the appropriate sample size is missing. Multiple modifications were implemented on the questionnaires; nevertheless, these alterations were not readily discernible. Which items were implicated? Who is responsible for validating the modifications during the survey process? The analysis pertaining to intra-group dynamics is missing.
Author Response

(The authors gave the same response as above.)

Round 2
Reviewer 1 Report
Comments and Suggestions for Authors
Manuscritp it's now ok.